

# Saudi radiology trainees' insights on safety and professionalism in the workplace

Khalid M. Alshamrani[1,2,3], Elaf K. Basalamah[1], Ghadah M. AlQahtani[1], Manar M. Alwah[1], Rahaf H. Almutairi[4], Walaa Alsharif[5], Awadia Gareeballah[5], Adnan AS Alahmadi[6], Shrooq T. Aldahery[7], Sultan A. Alshoabi[5] and Abdulaziz A. Qurashi[5]

[1] College of Applied Medical Sciences, King Saud bin Abdulaziz University for Health Sciences, Jeddah, Saudi Arabia
[2] King Abdullah International Medical Research Center, Jeddah, Saudi Arabia
[3] Ministry of the National Guard - Health Affairs, Jeddah, Saudi Arabia
[4] King Saud Medical City, Riyadh, Saudi Arabia
[5] Diagnostic Radiology Technology Department, College of Applied Medical Sciences, Taibah University, Al Madinah Al Munawwarah, Saudi Arabia
[6] Radiologic Sciences Department, Faculty of Applied Medical Sciences, King Abdulaziz University, Jeddah, Saudi Arabia
[7] Department of Applied Radiologic Technology, College of Applied Medical Sciences, University of Jeddah, Jeddah, Saudi Arabia

Corresponding author
Khalid M. Alshamrani,
alshamranik@ksau-hs.edu.sa

## ABSTRACT

**Introduction/Purpose:** In the radiology department, where advanced technologies and multidisciplinary collaboration are crucial, establishing a strong safety culture is particularly challenging. The present cross-sectional study examines the challenges of establishing a safety culture in radiology, focusing on how Saudi radiology trainees perceive and respond to safety and unprofessional conduct. It evaluates their willingness to voice concerns and the influencing factors, including workplace culture, potential patient risks, and demographics.

**Methods:** The present study surveyed Saudi radiology residents and interns at two tertiary hospitals using a validated questionnaire. A non-probability total population purposive sampling method was employed. Descriptive statistics, Mann-Whitney U test, and Kruskal-Wallis H test were used to analyze differences in willingness to speak up across demographic groups.

**Results:** Participants felt encouraged by colleagues to address patient safety and unprofessional behavior, with over 70% and 56% respectively agreeing. Residents demonstrated significantly greater support for raising concerns about safety and unprofessional conduct compared to interns (mean rank = 47.58 *vs.* 33.91, *p* = 0.009). Furthermore, residents expressed a stronger belief that speaking up leads to meaningful changes (mean rank = 46.24 *vs.* 35.36, *p* = 0.033) and reported observing others addressing these issues more frequently (mean rank = 46.98 *vs.* 34.56, *p* = 0.015). Trainees from different hospitals exhibited significantly varied perceptions regarding support from colleagues in addressing patient safety and unprofessional behavior (mean rank = KAMC 54.53 *vs.* KSMC 33.04, *p* < 0.0001), the perceived impact of raising concerns (mean rank = KAMC 50.50 *vs.* KSMC 35.41, *p* = 0.004), and the frequency of observing these concerns being addressed (mean rank = KAMC 55.28 *vs.* KSMC 32.60, *p* < 0.0001). Radiology trainees are particularly vigilant about unintentional breaches of sterile technique, often addressing these issues with nurses (66.7%).

**Conclusion:** The clinical environment supports safety concerns but less so for unprofessional behavior, with residents being more proactive. Promoting open communication in radiology requires leadership education, multifaceted strategies, alternative channels for concerns, and future research to assess and track cultural attitudes. The findings highlight the need to cultivate a supportive culture for speaking up in clinical settings, particularly in radiology, where trainee involvement can enhance patient safety and professional conduct. The present study lays the groundwork for future research and interventions to strengthen safety and professionalism among medical trainees in Saudi Arabia.

## INTRODUCTION

Building a safer healthcare environment requires an understanding that safety culture is a multifaceted and interconnected system of shared values that prioritize safety within clinical settings (*Slawomirski & Klazinga, 2022*; *Chau, 2024*). Safety culture is a dynamic framework composed of collective norms, and assumptions that guide behavior and decision-making in healthcare (*Bisbey et al., 2019*). It goes beyond simply implementing safety measures, focusing instead on establishing the foundational elements that sustain safe behaviors over time (*Bisbey et al., 2019*; *Tear et al., 2020*). By adopting such holistic approach, healthcare organizations can create a more resilient, adaptable, and effective safety culture, ultimately leading to improved patient safety and enhanced organizational performance (*Kilcullen et al., 2021*).

Fostering an environment where employees feel empowered to voice their concerns not only strengthens trust but also serves as a catalyst for enhanced performance, satisfaction, retention, productivity, innovation, and overall growth within the organization (*Detert & Burris, 2007*; *Detert & Treviño, 2010*; *Luff et al., 2021*). Leadership is crucial in cultivating a culture of safety within healthcare settings. Healthcare leaders are tasked with transforming the organization's vision and strategies into concrete safety measures (*Birken et al., 2018*; *Boutcher et al., 2022*). The critical role of managers emphasizes the need for clear communication and transparency to strengthen the safety culture. Leaders cultivate an atmosphere where psychological safety thrives, empowering staff to voice concerns and contribute ideas openly, free from the worry of repercussions (*O'donovan & Mcauliffe, 2020*). Such strategy is essential in developing a workplace where employees feel secure and supported (*Birken et al., 2018*; *O'donovan & Mcauliffe, 2020*; *Boutcher et al., 2022*).

Saudi Arabia's culture is characterized by high power distance, cultural tightness, and a high-context communication style. These cultural attributes indicate that hierarchy is deeply ingrained and accepted, social norms are strict with little tolerance for deviation, and communication relies heavily on implicit social context rather than explicit content. These cultural traits shape workplace behavior, particularly in terms of employee voice and

dissent (*Gelfand et al., 2011*; *Tear et al., 2020*). In high power distance settings like Saudi Arabia, power inequality is accepted, and individuals are more cautious about challenging authority. The tightness of the culture discourages deviation from established norms, further reducing tolerance for open dissent (*Harrington & Gelfand, 2014*). High-context communication emphasizes subtlety and sensitivity to social rules, especially when addressing superiors, which can discourage direct expression of disagreement (*Blair & Bligh, 2018*). Traditional studies of employee voice, largely based on Western contexts, may not fully apply to Saudi Arabia (*Lapinski & Rimal, 2005*). However, the high-context nature of the culture could allow alternative, indirect ways for employees to express their opinions. For instance, employees might suggest alternatives or use metaphors to voice dissent without being confrontational. This cultural dynamic highlights the need to explore culturally adapted methods for encouraging constructive workplace communication in Saudi Arabia.

In the medical field, experts emphasize that transparent communication about safety issues, such as adherence to hand-washing protocols and addressing unprofessional conduct, is crucial for cultivating robust safety cultures and achieving optimal outcomes (*Martinez et al., 2017*). Similarly, radiology patient safety leaders recognize that a strong organizational culture profoundly influences radiologic performance and outcomes, emphasizing the need for healthy team dynamics and respectful communication that empowers individuals to voice safety concerns or unprofessional conduct that could jeopardize patient safety (*Larson et al., 2015*; *Siewert & Hochman, 2015*; *Siewert et al., 2018*, *2019*). Radiology departments play a vital role in patient care by providing essential diagnostic and therapeutic services that rely on advanced technologies and collaboration across various disciplines (*Broder et al., 2018*). The complexity of these services, coupled with the fast-evolving technology and the large number of patient interactions, creates unique challenges in establishing and maintaining a strong safety culture (*Broder et al., 2018*; *Chau, 2024*).

Patient safety in radiology encompasses several critical aspects, including maintaining sterile techniques during interventional procedures, minimizing radiation exposure, ensuring accurate imaging interpretation, and adhering to infection prevention and control (IPC) protocols (*Abujudeh et al., 2017*; *Jimenez & Lewis, 2023*). Breaching sterile techniques can result in significant complications, such as nosocomial infections. For instance, case studies have reported bacterial infections in patients following breaches in hand hygiene by radiographers, and outbreaks linked to inadequate cleaning of radiology equipment (*Nihonyanagi et al., 2011*). Examples include joint infections after magnetic resonance arthrograms and bacterial contamination on computed radiography consoles and radiology workstations, often shared by multiple staff members (*Aso et al., 2011*; *Kim, Tyson & Mascola, 2013*; *Srivastava et al., 2021*). Infection risks extend to medical devices, as seen in an outbreak of acute hepatitis C among 12 patients exposed to contaminated saline flushes during CT imaging procedures (*Shteyer et al., 2019*). This incident highlights the critical need for strict adherence to sterile protocols, proper device handling, and routine disinfection of shared surfaces and equipment. Furthermore, radiologists play a pivotal role in patient safety by ensuring proper interpretation of images and timely

communication of findings (*Abujudeh et al., 2017*). The complexity of radiologic services and the integration of advanced technologies necessitate a robust safety culture to mitigate risks, improve outcomes, and prevent errors that could jeopardize patient care.

The level of empowerment that healthcare employees feel in voicing concerns about safety violations and unprofessional conduct in their work environment is becoming an important aspect of safety culture. Validated survey tools have been developed to measure this (*Martinez et al., 2015*; *Richard, Pfeiffer & Schwappach, 2021*), and research utilizing these tools has revealed that significant barriers to speaking up still exist in the healthcare setting (*Liao et al., 2014*; *Martinez et al., 2015*, *2017*; *Luff et al., 2021*). A 2018 study examined the culture surrounding the practice of speaking up about safety incidents within a major academic radiology department in the United States (*Siewert et al., 2018*). The study, which included 363 employees, found significant obstacles to reporting safety concerns, primarily due to the department's hierarchical structure. Similarly, a 2021 study surveyed a group of 58 radiology trainees across nine different training programs in the United States (*Luff et al., 2021*). The findings highlighted deficiencies in workplace cultures related to speaking up, especially in relation to unprofessional behavior and the influence of team hierarchy.

In Saudi Arabia, the Diagnostic Radiology Residency Training Program is a 4-year (R1 through R4) full-time residency conducted at accredited institutions. Residents undergo continuous and final evaluations through various examinations. The program follows a 4-week block format, comprising 13 rotations per training year and a total of 52 rotations throughout the program. Rotations cover a range of subspecialty areas, including Body Computed Tomography (CT) Scan, Neuroradiology, Ultrasound, Chest Imaging, Musculoskeletal (MSK) Imaging, Nuclear Medicine (NM)/Positron Emission Tomography (PET) Imaging, Pediatric Imaging, Breast Imaging, Body Magnetic Resonance Imaging (MRI), Fluoroscopy and Plain Films, Emergency Radiology/Plain Films, Interventional Radiology, Cardiac Imaging, Research/Quality Improvement (QI), Electives, and Core Skills. The program aims to develop the skills and interpretive abilities required for residents to become competent clinical diagnostic radiologists while fostering critical thinking and creativity as the foundation for sound clinical practice. Residents gain exposure to all aspects of radiology across various subspecialties and are encouraged to adopt a disciplined approach to medical problem-solving and decision-making (*Saudi Commission of Health Specialties, 2022*). In contrast, the Radiologic Technology Allied Health Internship is a 1-year (52-week) training program designed by individual universities for bachelor's degree students. Its goal is to develop professional practitioners who are clinically adaptable, competent, confident, and capable of critical thinking. Grounded in critical inquiry and evidence-based practice, the internship program promotes the acquisition of advanced clinical knowledge, skills, and behaviors essential for primary healthcare providers in a complex and ever-evolving healthcare environment.

In the present study, we adapted previously validated instrument tailored for medical and surgical trainees (*Martinez et al., 2015*), as well as radiology residents and fellows (*Luff et al., 2021*), to conduct a survey among radiology residents and interns at two tertiary hospitals in Saudi Arabia. Our study aimed to: (a) explore how these trainees perceive the

culture of their work environments in relation to voicing concerns about safety and unprofessional behavior (*i.e.*, negligent or willful deviations from the clinical standard of care), (b) evaluate their expected willingness to voice medical errors to radiology colleagues, and (c) identify the factors that influence this likelihood, including their perceptions of the speaking-up culture, the risk of patient harm associated with the error, and demographic factors.

## MATERIALS AND METHODS

### Participants and procedure

A descriptive cross-sectional study was carried out from January to February 2024, targeting radiology trainees at two tertiary hospitals in Saudi Arabia: King Abdulaziz Medical City (KAMC) in Jeddah at the Ministry of the National Guard-Health Affairs (MNG-HA), and King Saud Medical City (KSMC) in Riyadh. These hospitals were selected for their well-established radiology training programs, diverse patient populations, and comprehensive clinical environments, which enhance the relevance and generalizability of the study findings to the broader healthcare context in Saudi Arabia. KAMC in Jeddah is part of the Ministry of National Guard Health Affairs (MNG-HA), a government-funded healthcare system established in 1983. It is one of the leading medical institutions in Saudi Arabia, offering advanced healthcare services with a capacity exceeding 1,000 beds. The facility provides specialized medical care to the population of the Western Region and includes several distinguished centers, such as the Cardiology Center, the Neuroscience and Trauma Care Center, King Abdullah Specialized Children's Hospital, and Princess Noorah Oncology Center (*Alshamrani et al., 2023*, *2024b*, *2024a*). Similarly, KSMC is a tertiary care hospital under the Ministry of Health that has been serving Riyadh since 1956. With a capacity of 1,400 beds and a workforce of over 8,000 professionals, it is one of the largest healthcare institutions in the region. The medical complex encompasses several key hospitals, including the General Hospital, Pediatrics and Maternity Hospitals, the Dental Center, and King Fahad Charity Kidney Center (*Alshamrani et al., 2024a*). The study encompassed the entire cohort of 81 radiology trainees, consisting of 39 residents and 42 interns. Participants were selected through non-probability total population purposive sampling and were invited to participate *via* email and WhatsApp. The questionnaire was distributed online using Google Forms as the survey platform after securing the necessary permissions from the original source.

### Study measures

Radiology residents and interns completed established, previously validated scales designed to assess perceptions of safety culture and willingness to speak up about safety and professionalism concerns. These scales were originally validated in a study conducted across six major academic medical centers in the United States (*Martinez et al., 2015*), where an anonymous electronic survey was administered to residents. Confirmatory factor analysis identified two distinct, one-factor Speaking Up Climates (SUCs): one addressing patient safety concerns (SUC-Safe scale) and the other focusing on unprofessional behavior (SUC-Prof scale). Both scales demonstrated strong internal consistency, with

Cronbach's alpha values exceeding 0.70, confirming their reliability as measurement tools. These scales had been used in prior studies to assess internal medicine and surgery trainees, as well as radiology residents and fellows, regarding their perceptions of the workplace environment, specifically in relation to speaking up about traditional safety concerns and unprofessional behavior (*Martinez et al., 2015*, *2017*; *Luff et al., 2021*). To ensure relevance and clarity for our study's demographics, the questionnaires were reviewed by two radiologists and two senior radiology specialists with extensive experience in the field. To adapt the questionnaire to the Saudi context, specific modifications were made in the demographic section, including the addition of questions about participants' academic level, training hospital, and clinical experience. These adjustments were designed to reflect local cultural and clinical nuances, ensuring that the questionnaires were appropriately tailored, clear, and aligned with the study's objectives, thereby enhancing its reliability and ensuring the instrument's face validity.

The self-administered questionnaire was structured into three main sections. The first section included five demographic questions covering gender, age, academic level, training hospital, and clinical experience. The second section contained five domains and 10 items assessing participants' views on the level of support for voicing concerns about patient safety and unprofessional conduct in their clinical environments using a five-point Likert scale (1 = strongly disagree, 5 = strongly agree). The third section presented a hypothetical scenario, originally published and tailored for radiology, where a clinician accidentally compromises sterile technique during an imaging-guided central line placement (*Martinez et al., 2017*; *Luff et al., 2021*). The scenario reads: "You are working in the radiology suite when a clinician arrives to place a central catheter on a patient under radiographic guidance. The clinician sets up the supplies, prepares the patient, The clinician puts on a sterile gown and gloves, but then accidentally touches a nonsterile part of the ultrasound machine before proceeding to grab the catheter to place the line." Participants were then asked two questions: a) their likelihood of raising concerns about the clinician's breach of sterile technique and the likelihood that trainees would report this error to different staff members (attending radiologist, nurse, resident, or intern) using a five-point Likert scale (1 = not at all likely, 5 = completely likely); and b) their assessment of the potential risk to the patient in this scenario, also using a 5-point Likert scale (1 = very low, 5 = very high).

## Ethical consideration

The present study received approval from the local Institutional Review Board (IRB) of King Abdullah International Medical Research Center under protocol number SP23J/138/08. Participation was entirely voluntary, and written informed consent was obtained from all participants prior to completing the questionnaire. The consent form was embedded at the start of the Google Form survey, requiring participants to carefully read and confirm their agreement before proceeding. To maintain anonymity and confidentiality, all responses were kept anonymous, and the study followed the principles outlined in the Helsinki Declaration. The electronic survey tool generated a password-protected Microsoft Excel file, ensuring that no identifying information about participants was included.

## Statistical analyses

The statistical analysis was carried out through a systematic four-step approach. Initially, descriptive statistics were generated, including frequencies and percentages to summarize participant demographics and their responses to the questionnaire. Next, a weighted average was computed for the items within each domain as well as across all five domains in the second section of the questionnaire. Following this, the Shapiro–Wilk test was employed to assess the normality of the data. Finally, to explore differences in radiology trainees' willingness to speak up across various demographic groups, the Mann-Whitney U test and the Kruskal-Wallis H test were applied. The significance level was set at $\alpha < 0.05$, and all statistical analyses were performed using SPSS version 24.

## RESULTS

### Characteristics of the participants

Table 1 highlights the sociodemographic profile of 81 radiology trainees, with a gender distribution of 34 males (42%) and 47 females (58%). The age breakdown reveals that 43.2% of the trainees are between 20–24 years old, 28.4% are aged 25–29, and another 28.4% are 30 years or older. Academically, 51.9% are engaged in internship programs, while 48.1% are in residency. Most participants, 63.0%, have received their training at KSMC, while 37.0% were trained at KAMC. In terms of experience, 44.4% have less than 1 year, 30.9% have 1–5 years, and 24.7% have more than 5 years of experience.

### Perspectives to voice safety concerns and address unprofessional conduct: descriptive analysis

Table 2 presents the perspectives of radiology residents and interns on speaking up about safety concerns and unprofessional behavior. Participants felt a strong sense of encouragement from colleagues to address both traditional patient safety issues (mean: 3.68; with 70.4% agreeing or strongly agreeing) and unprofessional behavior (mean: 3.47; with 56.8% agreeing or strongly agreeing). Contrary to expectations, participants did not find it particularly difficult to speak up about these matters, with mean scores of 2.62 for safety concerns and 2.86 for unprofessional behavior, as 58.1% and 53.1%, respectively, disagreed or strongly disagreed that voicing these concerns was difficult. Trainees expressed a firm belief that voicing these concerns led to meaningful changes, with mean scores of 3.60 for safety issues and 3.54 for unprofessional behavior, and 55.6% agreeing or strongly agreeing in both cases. Radiology residents and interns also perceived the clinical culture as highly supportive of addressing safety concerns, with a mean of 3.59 and 60.5% agreeing or strongly agreeing, although the perceived support for tackling unprofessional behavior was slightly lower, with a mean of 3.26 and 46.9% in agreement. Additionally, participants noted frequent instances of others speaking up about safety concerns (mean: 3.59, with 59.2% agreeing or strongly agreeing) and unprofessional behaviors (mean: 3.48, with 50.7% agreeing or strongly agreeing).

**Table 1  Characteristics of the participants.**

| Variable | | Total sample = 81 | |
| --- | --- | --- | --- |
| | | *n* | % |
| Gender | Male | 34 | 42.0 |
| | Female | 47 | 58.0 |
| Age (years) | 20–24 | 35 | 43.2 |
| | 25–29 | 23 | 28.4 |
| | ≥30 | 23 | 28.4 |
| Academic level | Internship | 42 | 51.9 |
| | Residency | 39 | 48.1 |
| Training hospital | King Abdulaziz Medical City (KAMC) | 30 | 37.0 |
| | King Saud Medical City (KSMC) | 51 | 63.0 |
| Clinical experience (year) | <1 | 36 | 44.4 |
| | 1–5 | 25 | 30.9 |
| | >5 | 20 | 24.7 |

Note:
$-$ Percentage of Responses (%) $= \dfrac{\text{Number of Responses (n)}}{81} \times 100$

## Perspectives to voice safety concerns and address unprofessional conduct: inferential analysis

Tables 3 and 4 show the differences in radiology residents and interns' perspectives regarding the act of raising concerns about safety issues and unprofessional conduct across different demographic groups. The analysis revealed that residents exhibited significantly stronger overall support for raising concerns about safety and unprofessional conduct compared to interns ($p = 0.009$). Notably, residents were more likely to believe that speaking up led to meaningful changes ($p = 0.033$) and reported more frequent observations of others addressing these issues ($p = 0.015$) than their intern counterparts. The observed differences between residents and interns likely stem from variations in training exposure and professional responsibility. Residents, having more advanced training and clinical experience, are likely more confident in identifying and addressing safety concerns and unprofessional conduct. Additionally, their increased exposure to clinical scenarios and greater responsibility in patient care may contribute to a stronger belief in the impact of speaking up and more frequent observations of these behaviors among colleagues. Additionally, radiology residents and interns trained at KAMC showed significantly greater overall support for addressing safety concerns and unprofessional behavior compared to those trained at KSMC ($p < 0.0001$). KAMC trainees felt a stronger sense of encouragement from colleagues to address traditional patient safety issues and unprofessional behavior ($p < 0.0001$), were more likely to believe that raising these concerns resulted in meaningful changes ($p = 0.004$), and observed others addressing such issues more frequently ($p < 0.0001$) than their counterparts at KSMC. The differences in perceptions between trainees at KAMC and KSMC may be influenced by institutional factors such as organizational culture, leadership styles, and emphasis on patient safety and

**Table 2 Radiology trainees' perspectives on speaking up: comparing views on safety concerns and unprofessional behavior.**

| Domain/Item | Strongly disagree | Disagree | Natural | Agree | Strongly agree | Mean | σ | 95% CI | Overall perception level |
|---|---|---|---|---|---|---|---|---|---|
| **Colleague encouragement:** | Weighted average = 3.57 | | | | | | | | |
| a) I am encouraged by my colleagues to speak up about traditional patient safety concerns. | 4 (5%) | 10 (12.3%) | 10 (12.3%) | 41 (50.6%) | 16 (19.8%) | 3.68 | 1.08 | 3.44–3.92 | High |
| b) I am encouraged by my colleagues to speak up about unprofessional behavior. | 4 (5%) | 14 (17.3%) | 17 (20.9%) | 32 (39.5%) | 14 (17.3%) | 3.47 | 1.12 | 3.22–3.72 | |
| **Difficulty speaking up:** | Weighted average = 2.74 | | | | | | | | |
| a) In my clinical area, it is difficult to speak up if I have traditional patient safety concerns. | 6 (7.4%) | 41 (50.7%) | 13 (16.0%) | 20 (24.7%) | 1 (1.2%) | 2.62 | 0.98 | 2.40–2.83 | Moderate |
| b) In my clinical area, it is difficult to speak up if I observe unprofessional behavior. | 12 (14.8%) | 31 (38.3%) | 13 (16.1%) | 21 (25.9%) | 4 (4.9%) | 2.86 | 1.16 | 2.42–2.94 | |
| **Meaningful change:** | Weighted average = 3.57 | | | | | | | | |
| a) Speaking up about traditional patient safety concerns results in meaningful change in my clinical area. | 1 (1.2%) | 5 (6.2%) | 30 (37.0%) | 34 (42.0%) | 11 (13.6%) | 3.60 | 0.84 | 3.42–3.79 | High |
| b) Speaking up about unprofessional behavior results in meaningful change in my clinical area. | 2 (2.5%) | 10 (12.3%) | 24 (29.6%) | 32 (39.5%) | 13 (16.1%) | 3.54 | 0.98 | 3.32–3.76 | |
| **Clinical culture:** | Weighted average = 3.43 | | | | | | | | |
| a) The culture in my clinical area makes it easy to speak up about traditional patient safety concerns that do not involve me or my patients. | 0 (0%) | 10 (12.3%) | 22 (27.2%) | 40 (49.4%) | 9 (11.1%) | 3.59 | 0.84 | 3.41–3.78 | Moderate |
| b) The culture in my clinical area makes it easy to speak up about unprofessional behavior that does not involve me or my patients. | 4 (4.9%) | 13 (16.1%) | 26 (32.1%) | 34 (42.0%) | 4 (4.9%) | 3.26 | 0.95 | 3.05–3.47 | |
| **Observe others speaking up:** | Weighted average = 3.54 | | | | | | | | |
| a) In my clinical area, I observe others speaking up about traditional patient safety concerns even if they are not directly involved in the patient's care. | 0 (0%) | 11 (13.6%) | 22 (27.2%) | 37 (45.6%) | 11 (13.6%) | 3.59 | 0.89 | 3.40–3.79 | High |
| b) In my clinical area, I observe others speaking up about unprofessional behavior even if they are not directly involved in the patient's care. | 1 (1.2%) | 10 (12.3%) | 29 (35.8%) | 31 (38.4%) | 10 (12.3%) | 3.48 | 0.91 | 3.28–3.68 | |
| **Overall weighted average =** | 3.35 | | | | | | | | |

Notes:

Percentage of Responses (%) $= \dfrac{\text{Number of Responses (n)}}{81} \times 100$

σ = Standard deviation.

CI = Confidence interval.

The levels of the mean scores on five-point Likert Scale: <1.5 = Very Low; 1.5–2.5 = Low; 2.5–3.5 = Moderate; 3.5–4.5 = High; 4.5–5 = Very High.

professionalism in training programs. KAMC's environment, which appears to foster a stronger culture of encouragement and accountability, may explain why its trainees reported greater support and more frequent observations of addressing safety concerns and unprofessional behavior. Additionally, differences in the perceived openness of communication and support within the hierarchical structures of each institution may play

**Table 3 Radiology trainees' perspectives on speaking up: analyzing inferential statistics by gender, academic level, and training hospital.**

| Domain | Gender | | | |
| --- | --- | --- | --- | --- |
| | Mean rank | | Mann-Whitney U | P-value |
| | Male | Female | | |
| Colleague encouragement | 43.62 | 39.11 | 710.0 | 0.338 |
| Difficulty speaking up | 42.19 | 40.14 | 758.5 | 0.692 |
| Meaningful change | 38.93 | 42.5 | 896.5 | 0.490 |
| Clinical culture | 43.31 | 39.33 | 720.5 | 0.438 |
| Observe others speaking up | 39.72 | 41.93 | 842.5 | 0.670 |
| Total | 41.71 | 40.49 | 775.0 | 0.817 |
| Domain | Academic level | | | |
| | Mean rank | | Mann-Whitney U | P-value |
| | Radiolgy residency | Internship | | |
| Colleague encouragement | 44.51 | 77.21 | 671.0 | 0.157 |
| Difficulty speaking up | 41.62 | 40.33 | 793.0 | 0.802 |
| Meaningful change | 46.24 | 35.36 | 599.0 | 0.033* |
| Clinical culture | 45.63 | 36.01 | 624.5 | 0.058 |
| Observe others speaking up | 46.98 | 34.56 | 568.0 | 0.015* |
| Total | 47.58 | 33.91 | 542.5 | 0.009* |
| Domain | Training hospital | | | |
| | Mean rank | | Mann-Whitney U | P-value |
| | KAMC | KSMC | | |
| Colleague encouragement | 54.53 | 33.04 | 359.00 | <0.0001* |
| Difficulty speaking up | 35.17 | 44.43 | 940.00 | 0.080 |
| Meaningful change | 50.50 | 35.41 | 480.00 | 0.004* |
| Clinical culture | 46.90 | 37.53 | 588.00 | 0.074 |
| Observe others speaking up | 55.28 | 32.60 | 336.5 | <0.0001* |
| Total | 53.97 | 33.37 | 376.00 | <0.0001* |

Notes:
* Significance.
KAMC, King Abdulaziz Medical City; KSMC, King Saud Medical City.

a significant role. Gender, however, did not significantly influence the trainees' willingness to voice safety concerns or address unprofessional conduct ($p = 0.817$) (Table 3). Furthemore, individuals with less than 1 year of work experience and those aged 20–24 demonstrated significantly greater overall support for addressing safety concerns and unprofessional behavior compared to their counterparts with over 5 years of experience and those aged 30 or older ($p = 0.026$; $p = 0.017$, respectively). These younger and less experienced trainees were also more likely to observe others addressing these issues more frequently ($p = 0.003$; $p = 0.001$, respectively) than their more experienced, older peers (Table 4).

**Table 4 Radiology trainees' perspectives on speaking up: analyzing inferential statistics by years of experience and age.**

**Years of experience**

| Domain | Mean rank | | | Kruskal-Wallis H | P-value |
|---|---|---|---|---|---|
| | <1 | 1–5 | >5 | | |
| Colleague encouragement | 46.74 | 37.88 | 34.58 | 4.191 | 0.123 |
| Difficulty speaking up | 39.79 | 40.56 | 43.72 | 0.389 | 0.823 |
| Meaningful change | 46.86 | 37.56 | 34.75 | 4.379 | 0.112 |
| Clinical culture | 45.47 | 37.38 | 37.48 | 2.493 | 0.288 |
| Observe others speaking up | 48.72 | 41.08 | 27.00 | 11.475 | 0.003* |
| Total | 48.08 | 38.94 | 30.82 | 7.273 | 0.026* |

**Age (Years)**

| Domain | Mean rank | | | Kruskal-Wallis H | P-value |
|---|---|---|---|---|---|
| | 20–24 | 25–29 | ≥ 30 | | |
| Colleague encouragement | 47.43 | 37.67 | 34.54 | 4.946 | 0.084 |
| Difficulty speaking up | 38.9 | 41.65 | 43.54 | 0.591 | 0.744 |
| Meaningful change | 47.94 | 35.87 | 35.57 | 5.624 | 0.060 |
| Clinical culture | 45.53 | 38.78 | 36.33 | 2.564 | 0.277 |
| Observe others speaking up | 49.00 | 43.91 | 25.91 | 14.508 | 0.001* |
| Total | 48.3 | 40.54 | 30.35 | 8.181 | 0.017* |

**Note:**
* Significance.

**Table 5 Likelihood of addressing a medical error in the hypothetical scenario: odds of speaking up.**

| Item | Not at all likely n (%) | Unlikely | Natural | Likely | Completely likely | Mean ± SD | Finding |
|---|---|---|---|---|---|---|---|
| Nurse | 3 (3.7%) | 10 (12.3%) | 14 (17.3%) | 20 (24.7%) | 34 (42%) | 3.89 ± 1.19 | Likely |
| Intern | 9 (11.1%) | 9 (11.1%) | 15 (18.5%) | 16 (19.8%) | 32 (39.5%) | 3.65 ± 1.38 | Less likely |
| Resident | 3 (3.7%) | 7 (8.6%) | 26 (32.0%) | 19 (23.7%) | 26 (32.0%) | 3.72 ± 1.12 | Likely |
| Attending radiologist | 7 (8.6%) | 12 (14.8%) | 16 (19.8%) | 23 (28.4%) | 23 (28.4%) | 3.53 ± 1.28 | Less likely |
| Weighted average = 3.66 | | | | | | | |

**Notes:**
$$\text{Percentage of Responses (\%)} = \frac{\text{Number of Responses (n)}}{81} \times 100$$
SD = Standard deviation.

## Addressing medical errors within the clinical hierarchy

Table 5 presents the factors influencing participants' likelihood of speaking up about a hypothetical scinario of unintentional breach of sterile technique by a clinician. Radiology residents and interns were likely or completely likely to address the issue with a nurse (66.7%), followed by an intern (59.3%), an attending radiologist (56.8%), and a resident (55.7%). Additionally, 54.32% of the 81 radiology trainees perceived this error as having a high or very high potential for patient harm (Fig. 1). Participants showed a preference for raising issues with individuals perceived as more approachable, such as nurses and peers, rather than higher-ranking staff like attending radiologists. These communication

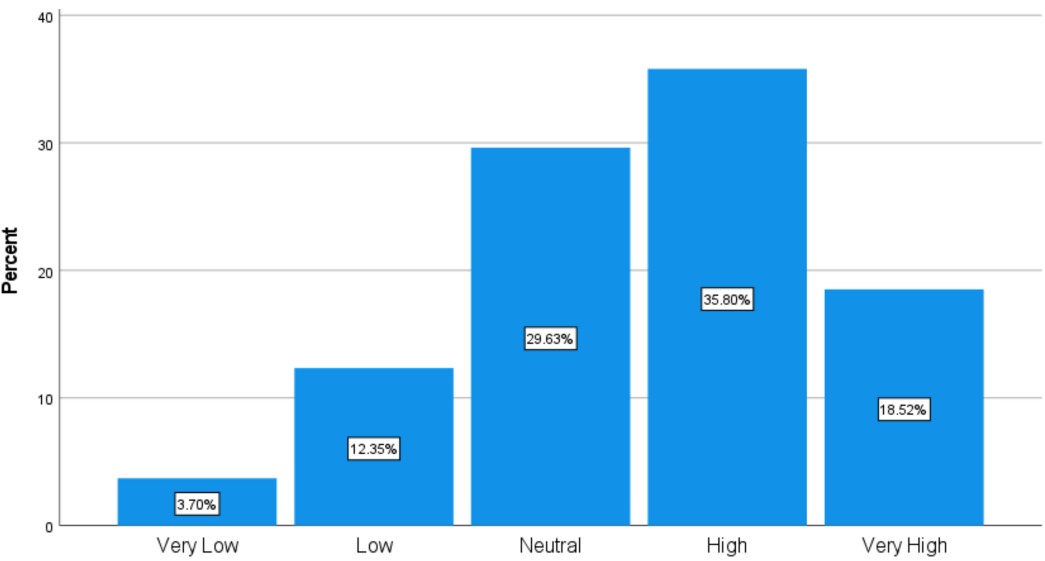

**Figure 1 Risk of patient harm due to unintentional breach of sterile technique by clinician.**

patterns suggests that the hierarchical dynamics within clinical settings may affect communication pathways. Additionally, the high percentage of trainees perceiving the error as having significant potential for patient harm underscores the importance of fostering an environment where trainees feel empowered to escalate concerns regardless of rank.

## DISCUSSION

### Statement of principal findings

This cross-sectional study explores how Saudi radiology trainees perceive the culture of addressing safety and unprofessional behavior, evaluates their willingness to report medical errors, and identifies factors influencing these reporting behaviors, including speaking-up culture, potential patient harm, and demographic factors. To the best of our knowledge, this is the first study to specifically examine these factors among radiology residents and interns, both within Saudi Arabia and across other medical disciplines in the country. The present study reveals several key findings: firstly, radiology trainees, including both residents and interns, generally feel encouraged by their colleagues to address issues related to safety and unprofessional behavior. Importantly, more than half believe that voicing their concerns leads to meaningful changes. Secondly, the clinical environment is perceived as supportive of addressing safety concerns, although there is slightly less perceived support when it comes to tackling unprofessional behavior. Third, radiology residents are notably more proactive and supportive in raising concerns about safety and unprofessional behavior compared to interns. Fourth, trainees, particularly those at KAMC and those with less than 1 year of experience, show a significantly stronger commitment to address safety concerns and unprofessional behavior than their more experienced colleagues and those trained at KSMC. Fifth, radiology trainees are

particularly vigilant about unintentional breaches of sterile technique, often addressing these issues with nurses. Over half of the trainees view such errors as having a high potential for patient harm.

## Interpretation within the context of the wider literature

Our study uniquely addresses a critical gap in the current literature regarding the culture of speaking up within the field of radiology, with a specific focus on the Saudi Arabian context. To date, there has been no research exploring the dynamics of this issue within Saudi Arabia, nor any studies that examine whether these dynamics are consistent across different countries. Our research seeks to understand how these conditions impact radiology trainees (*i.e.*, residents and interns), the influence of safety event severity on the willingness to speak up, and the role of hierarchical structures in shaping individuals' willingness to voice concerns related to both traditional safety issues, such as non-sterile techniques, and unprofessional behavior. Given the well-documented links between a culture of respect and safety, as well as the connection between unsafe or disrespectful behavior and malpractice (*Leape et al., 2012*; *Webb et al., 2016*; *Cooper et al., 2017*, *2019*; *Riskin et al., 2015*), our study is significant in contributing valuable insights into these critical aspects within radiology.

Our research indicates that radiology residents and interns recognize challenges in voicing concerns within their clinical settings, particularly when it comes to addressing unprofessional behavior. This observation aligns with the results of *Luff et al. (2021)*, who conducted a similar study involving 58 radiology trainees, as well as with the findings of *Martinez et al. (2017)*, who surveyed a large group of 1,800 medical and surgical interns and residents using the same five-domain, ten-item tool. Both studies also indicated that participants were less likely to report instances of unprofessional behavior than they were to raise concerns about safety issues. These findings are significant, as unprofessional behavior can undermine team effectiveness, disrupt communication, and create a culture of mistrust, all of which negatively impact patient safety. Addressing unprofessional conduct is thus essential for fostering a collaborative clinical environment and ensuring optimal healthcare outcomes. (*Leape et al., 2012*; *Riskin et al., 2015*, *2019*; *Cooper et al., 2017*, *2019*; *Dixon-Woods et al., 2018*; *Lagoo et al., 2018*).

Our study revealed that radiology residents and interns were more likely to raise safety concerns and address unprofessional behavior with a fellow nurse than with an attending radiologist or another resident or intern. This communication pattern suggests that hierarchical structures may discourage trainees from speaking up. Such findings are consistent with previous research, which has highlighted the widespread presence of hierarchical barriers that hinder open communication about safety concerns in various healthcare settings (*Martinez et al., 2017*; *Luff et al., 2021*). Previous studies have positioned radiology within a pervasive cultural context where clinical staff often feel limited in their ability to discuss safety issues across different levels of authority (*Okuyama, Wagner & Bijnen, 2014*; *Martinez et al., 2017*; *Luff et al., 2021*). Our study highlighted the significant role of workplace culture in either promoting or discouraging speaking-up behavior among radiology residents and interns, echoing similar results found in other

clinical groups (*Siewert et al., 2019*). The data reinforces that cultures which encourage open communication foster not only enhance patient safety but also contribute to the well-being of trainees by offering psychologically safe environments for those who are vulnerable within a hierarchical structure (*Okuyama, Wagner & Bijnen, 2014*; *Osseo-Asare et al., 2018*). In contrast, environments that suppress open dialogue can result in moral distress, burnout, and emotional (*Frazier et al., 2017*; *Newman, Donohue & Eva, 2017*; *Osseo-Asare et al., 2018*). Leadership training programs could play a pivotal role in addressing hierarchical barriers and fostering a culture of open communication. Such programs could focus on empowering both trainees and senior staff to engage in constructive dialogues, promote psychological safety, and emphasize the importance of addressing safety concerns and unprofessional behavior at all levels. Integrating leadership training into radiology residency programs could contribute to breaking down hierarchical barriers, improving patient safety, and enhancing the overall well-being of trainees.

Despite the similar sizes of KAMC and KSMC, trainees at KAMC reported a stronger commitment to addressing safety concerns and unprofessional behavior compared to their counterparts at KSMC. These findings suggests that the training environment is a critical factor in shaping the willingness to raise concerns, emphasizing the importance of cultivating a supportive culture in medical training programs to encourage open communication on safety and professionalism (*Mistri, Badge & Shahu, 2023*; *Alsahli et al., 2024*). Additioally, hospital size is a key factor to consider, as research highlights its influence on managerial practices in healthcare organizations (*El-Jardali et al., 2014*). Larger institutions often have an advantage in meeting accreditation standards, such as patient safety culture assessments, due to their ability to distribute costs more effectively within their overall budgets (*El-Jardali et al., 2008*). Studies also suggest that larger hospitals tend to achieve better quality outcomes following accreditation. However, smaller organizations benefit from a more cohesive culture and shared values, which can enhance communication and teamwork (*Andres et al., 2019*). In contrast, larger hospitals tend to have more hierarchical and bureaucratic structures, which can make implementing quality improvement initiatives more challenging (*Alasmari et al., 2021*). This, in turn, may impact employees' sense of connection to the organization and their overall performance (*Lucifora, 2023*). Evidence also suggests that smaller hospitals (typically under 100 beds) demonstrate stronger formal leadership engagement in patient safety events. This closer leadership presence at the frontline contributes to improved patient safety behaviors, particularly in smaller hospitals where the financial burden of safety programs is more significant (*Ginsburg et al., 2010*).

Moreover, radiology residents demonstrate a notably higher level of proactivity and support in addressing safety concerns and unprofessional behavior compared to interns. This increased engagement suggests that the extended experience and training that residents receive—comprising 7 years of medical school followed by a 5-year residency—fosters a stronger sense of responsibility and confidence in managing critical issues within the clinical setting. In contrast, interns typically have only 4 years of radiological sciences

education before beginning their internships, which may contribute to their more limited engagement in these areas.

## Strengths and limitations

One of the strengths of our study is that we conducted a survey among trainees from two major tertiary hospitals, providing a robust data set. To the best of our knowledge, this is the first study to explore the culture of speaking up among radiology residents and interns in Saudi Arabia. Our findings contribute to the existing literature by not only assessing traditional safety concerns but also examining residents' and interns' experiences and attitudes toward addressing unprofessional behavior. Nonetheless, our study does have limitations. While purposive sampling enables a deeper and more detailed investigation, enriching the study's overall insights, it also introduces inherent bias, which limits the ability to generalize the findings to a wider population and potentially affects the reliability of the study's findings. Furthermore, the patient safety scenario included in our questionnaire was hypothetical, which represents a limitation of the study, as it may not fully reflect participants' behavior in real-world clinical settings. Self-reported intentions in hypothetical scenarios may differ from actual behavior due to contextual factors such as stress, time constraints, or hierarchical pressures. However, these scenarios provide valuable insights into participants' attitudes, perceptions, and perceived barriers, which are critical for informing future research and the development of targeted interventions. Additionally, the survey did not include a scenario on unprofessional behavior, limiting the ability to assess how likely radiology residents and interns are to speak up about such issues. However, this limitation does not detract from our primary findings regarding residents and interns' perceptions of safety culture *vs.* unprofessional behavior. Moreover, the study did not clearly define the specific types of unprofessional behavior referenced in the questions. As a result, participants might have interpreted unprofessional behavior differently, leading to varied perceptions of its severity. Additionally, radiology residents and interns' understanding of what constitutes unprofessional behavior may differ depending on the context and the way the questions were presented (*Wong & Ginsburg, 2017*). Unprofessional behavior can manifest in various ways, ranging from overt harassment and misconduct to more subtle acts of unreasonable demands, incivility, disrespect, and bullying (*Dixon-Woods et al., 2018*). Although these behaviors have been acknowledged in the radiology literature, their frequency and associated impacts are not yet fully understood (*Rawson et al., 2013*; *Brown et al., 2014*). Another limitation is the potential Hawthorne effect, a phenomenon in which individuals alter their behavior in response to being observed or knowing they are part of a study. In the context of our study, which compares radiology trainees at KAMC and KSMC in addressing patient safety concerns and unprofessional behavior, participants may have responded in a manner that aligns with socially desirable norms rather than their actual perceptions or experiences. This could have resulted in an overestimation of their reported engagement in addressing these issues. To mitigate this effect, a follow-up study incorporating independent observations of trainee behaviors in clinical settings could provide a more objective assessment of institutional differences. Additionally, future research may explore

alternative methodologies, such as qualitative interviews or focus groups, to gain deeper insights into trainees' perspectives.

### Implications for policy, practice and research

Our findings highlight an increasing awareness within medicine and radiology of the need for systemic reforms to cultivate professional cultures where staff feel confident in voicing safety concerns and addressing unprofessional behavior (*Pian-Smith et al., 2009*; *Kruskal et al., 2019*; *Siewert et al., 2019*; *Luff et al., 2021*). Crucial measures involve educating leadership about the detrimental effects of unsupportive environments on both patient care and staff well-being, and creating strategies to remove obstacles to open dialogue (*Profit et al., 2014*; *Etchegaray et al., 2017*; *Dixon-Woods et al., 2018*). Some organizations have effectively employed simulation exercises and educational programs to equip staff with the skills to address safety concerns constructively (*Pian-Smith et al., 2009*; *Okuyama, Wagner & Bijnen, 2014*; *Dixon-Woods et al., 2018*). Simulated scenarios provide a safe and controlled environment for trainees to practice raising safety concerns, improve communication skills, and navigate hierarchical dynamics, ultimately fostering a culture of openness and psychological safety within healthcare teams. Leadership must exemplify these behaviors to successfully build and maintain a culture that encourages open communication (*Etchegaray et al., 2017*). Promoting a culture of speaking up in radiology likely demands a multifaceted approach, as training alone may be insufficient (*Raemer et al., 2016*). Suggestions include establishing alternative channels for raising concerns, ensuring diverse voices are heard, and creating informal settings where hierarchical structures do not inhibit honest communication (*Luff et al., 2021*). Anonymous online reporting tools can also empower staff to express concerns (*Webb et al., 2016*; *Martinez et al., 2017*; *Siewert et al., 2019*). Tackling these barriers, alongside issues of unprofessional behavior, can provide valuable insights for future research and interventions aimed at enhancing open communication within radiology departments (*Siewert et al., 2019*).

Future initiatives should involve conducting an extensive national survey across Saudi Arabia that explore the cultural attitudes toward addressing safety concerns and unprofessional behavior within the field of radiology. This survey should aim to capture real clinical incidents and include a diverse participant group to ensure comprehensive insights. The instrument adapted for this study can serve as a valuable tool for longitudinal assessments, allowing researchers to measure cultural shifts in radiology over time, particularly following the introduction of interventions designed to encourage more open communication and reporting of safety issues. Additionally, future research may incorporate scenarios depicting unprofessional behavior to provide a more comprehensive assessment of trainees' insights and responses. This approach would further enrich the study's findings and better address its overarching objectives.

## CONCLUSIONS

The present cross-sectional study provides valuable insights into how Saudi radiology trainees perceive the culture of voicing concerns about safety and unprofessional conduct, and their propensity to report medical errors, and the factors influencing these behaviors.

Radiology trainees, both residents and interns, feel encouraged by peers to address safety and unprofessional behavior, with over half believing their concerns lead to change. While the clinical environment supports safety concerns, there's less support for unprofessional behavior. Residents, particularly at KAMC, are more proactive than interns, likely due to their extensive training. Trainees also demonstrate strong awareness of potential patient harm, especially regarding sterile technique breaches. Overall, these findings highlight the importance of fostering a supportive culture for speaking up within clinical environments, particularly in radiology, where the proactive involvement of trainees can significantly enhance patient safety and professional conduct. This study serves as a foundational step for future research and interventions aimed at strengthening the culture of safety and professionalism among medical trainees in Saudi Arabia. To foster a culture of safety and professionalism in radiology training, the establishment of anonymous reporting systems and informal feedback channels is recommended to mitigate hierarchical barriers. Additionally, integrating simulation-based training and leadership development programs into radiology curricula can equip trainees with the skills and confidence needed to address safety concerns and unprofessional behavior effectively. Educators and policy-makers should prioritize creating psychologically safe environments that encourage open dialogue, empowering trainees to contribute to a culture that prioritizes patient safety and professional integrity.

## ABBREVIATIONS

| | |
|---|---|
| **BLS** | Basic Life Support |
| **CPR** | Cardiopulmonary Resuscitation |
| **KSAU-HS** | King Saud bin Abdulaziz University for Health Sciences |
| **UJ** | University of Jeddah |
| **TU** | Taibah University |
| **AHA** | American Heart Association |
| **SCFHS** | Saudi Commission for Health Specialties |
| **GPA** | Grade Point Average |
| **AED** | Automated External Defibrillator |
| **IRB** | Institutional Review Board |

## ACKNOWLEDGEMENTS

We thank all residents and interns for participating in the study.

### Funding

The authors received no funding for this work.

### Competing Interests

The authors declare that they have no competing interests.

## Author Contributions

- Khalid M. Alshamrani conceived and designed the experiments, performed the experiments, analyzed the data, prepared figures and/or tables, authored or reviewed drafts of the article, and approved the final draft.
- Elaf K. Basalamah performed the experiments, authored or reviewed drafts of the article, and approved the final draft.
- Ghadah M. AlQahtani performed the experiments, authored or reviewed drafts of the article, and approved the final draft.
- Manar M. Alwah performed the experiments, authored or reviewed drafts of the article, and approved the final draft.
- Rahaf H. Almutairi performed the experiments, authored or reviewed drafts of the article, and approved the final draft.
- Walaa Alsharif performed the experiments, analyzed the data, prepared figures and/or tables, authored or reviewed drafts of the article, and approved the final draft.
- Awadia Gareeballah performed the experiments, analyzed the data, prepared figures and/or tables, authored or reviewed drafts of the article, and approved the final draft.
- Adnan AS Alahmadi performed the experiments, authored or reviewed drafts of the article, and approved the final draft.
- Shrooq T. Aldahery performed the experiments, authored or reviewed drafts of the article, and approved the final draft.
- Sultan A. Alshoabi performed the experiments, authored or reviewed drafts of the article, and approved the final draft.
- Abdulaziz A. Qurashi performed the experiments, authored or reviewed drafts of the article, and approved the final draft.

## Human Ethics

The following information was supplied relating to ethical approvals (*i.e.*, approving body and any reference numbers):

King Abdullah International Medical Research Center granted approval for this study, designated as Study Number: SP23J/138/08.

## Data Availability

The raw measurements are available in the Supplemental File.

## Supplemental Information

Supplemental information for this article can be found online at http://dx.doi.org/10.7717/peerj.19257#supplemental-information.

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
