# Peer review of "Saudi radiology trainees’ insights on safety and professionalism in the workplace"

_PeerJ, doi:10.7717/peerj.19257_

## Round 0.1 · original submission · Minor Revisions

Please respond to all the comments of the 3 reviewers

·

Basic reporting

Please see the detailed reviewer report

Experimental design

Please see the detailed reviewer report

Validity of the findings

Please see the detailed reviewer report

Additional comments

Please see the detailed reviewer report

Reviewer 2 ·

Basic reporting

This is a promising paper, however does not provide the reader adequate or supporting literature about the precise focus and context. In the introduction, there is discussion of safety in general. However, there is a need for further context of what is safety in radiology and the radiologist role in maintaining safety. For example, it would be in this section that there would be an initial explanation that breaching sterile technique is an issue, as well as brief description of other top issues. In sum, what is meant by safety for this specialty and, just as important, what are the consequences? As an example, X patients per year receiving oncological radiation experience nosocomial infections, with citation. If such a statistic specific to radiology is not available, that is fine, although it would be ideal. The general goal is simply to ensure the reader understand the clinical problem being addressed in the paper.

One additional reason this is very important is that the term “unprofessional behavior” (used in the instrument) and the term “unprofessional conduct” (used in the manuscript) have several different connotations both in the paper and in clinical practice, and it is very possible a reader will be confused about the focus of the paper. Is this referring to the 1. Clinical outcomes focused issues such as sterile technique (as in the case study example of the instrument) above or 2. Harassment and bullying (line 344)? If the second component is going to be included, then it also requires appropriate explanation and context. Or is the “unprofessional behavior” defined as negligent or willful deviations from the clinical standard of care that appears to be the main focus of the paper? If there is a hypothesis that professionals of lower rank may be fearful of reporting clinical concerns they observe precisely because they may experience “unprofessional behavior” (still needs to be defined), then this should be explained more clearly in the introduction.

Overall, the technical writing of the paper is well done, I have just a few minor suggestions. Regarding the title, I do not recommend the Voices of Change component, as that would seem to imply a Qualitative component to the piece – this should probably be omitted. The abstract should state immediately it is a cross-sectional study. Please read over the headings, as they do not seem to be accurately formatted. Several in-text citations that are included in the same parentheses are not in alphabetical order. There are several awkward uses of the pronoun “this” (avoid using as pronoun and replace with a noun).

Experimental design

Regarding the methodology, the authors state several times that the measure has been previously validated, but a Cronbach’s alpha is not reported. If available this should be included, with a description of the previous setting. Good description of technical aspects of how study was carried out and the data analysis is appropriate.

Validity of the findings

An issue which presents a strong concern is the reporting of contrasting findings between the two hospitals. This runs counter to standard practice, and does not provide meaningful information to the many readers who do not know of any significant differences between the facilities. What would be far more useful is to provide informative descriptors for the two hospitals. Example: Hospital A is an urban, teaching hospital. Hospital B is a smaller, suburban hospital. (These are examples, of course, the authors should provide accurate, informed descriptions). Then, report the differences in results according to the hospital characteristics and in the discussion, you might then hypothesize about any possible relationships, and you can compare to existing literature. “We found that safety practices at the teaching hospital were significantly different from those at the smaller hospital, and this concurred/varied with the findings of XYZ other authors et al….”

The authors should also acknowledge as limitations the potential for the Hawthorne effect to influence participants to increase the potential for them to respond in a socially desirable manner, especially one such as this one. Perhaps follow up study can include independent observation.

Additional comments

Thank you for the opportunity to review this manuscript. This paper is a cross-sectional study of self-reported safety attitudes among radiology interns and residents. The investigation has some promising elements, however also has some areas that I would strongly recommend be addressed before publication. Best wishes on this promising line of inquiry.

·

Basic reporting

The writing of the manuscript is smooth and clear with academic English expression and is easy to understand. The introduction of the research background is sufficient, and the reference is selected and cited properly. The document structure meets the requirements of Peer J, pictures and tables are with clear legent and description, and the raw data is completely supplied.

Experimental design

This manuscript titled “Voices of change: Saudi radiology trainees' insights on safety
and professionalism in the workplace” carried a descriptive cross-sectional study in Saudi Arabia about radiology residents and interns on speaking up about safety concerns and unprofessional behavior. The research design is reasonable and the results are credible, which has significance for establishing and improving hospital safety culture

Validity of the findings

no comment

Additional comments

minor tips:
1.The INTRODUCTION section which contains 6 paragraphs seems too lengthy and some of the content overlaps with the content of the DISCUSSION section. I suggest to make an appropriately simplify like: move paragraphs 4 and 5(line 113-137), especially paragraph 5(line 126-137) to the DISCUSSION section.
2.As for the data presentation of the results of the two questions in section 3 of the questionnaire, the author shows them by Table 5 and Figure 1. I feel both of then be demonstrated by figures(histogram) is more intuitive and consistent。
3.I prefer a structural ABSTRACT with sections of introduction/purpose, methods, results and conclusion

---

## Round 0.2 · accepted · Accept

All reviewers and I are satisfied with the revision and agree that the manuscript should now be published in its current form.

·

Basic reporting

Clear and unambiguous, professional English used throughout.

Literature references, sufficient field background/context provided.

Professional article structure, figures, tables. Raw data shared.

Self-contained with relevant results to hypotheses.

Experimental design

Original primary research within Aims and Scope of the journal.
Research question well defined, relevant & meaningful. It is stated how research fills an identified knowledge gap.

Rigorous investigation performed to a high technical & ethical standard.

Methods described with sufficient detail & information to replicate.

Validity of the findings

Impact and novelty not assessed. Meaningful replication encouraged where rationale & benefit to literature is clearly stated.

All underlying data have been provided; they are robust, statistically sound, & controlled.

Conclusions are well stated, linked to original research question & limited to supporting results.

Additional comments

I commend the authors for their work and recommend this paper for publication.

Reviewer 2 ·

Basic reporting

This cross-sectional study had been extensively revised and is much stronger with the additional information. The additional literature to demonstrate the significance of the reseach question is excellent. The one suggestion I present for the authors' consideration is that based on this content, it seems the main topic of the paper is safety communication.

Experimental design

The previous remarks about experimental design have been fully addressed.

Validity of the findings

Previous remarks about validity of the findings have been fully addressed.

Additional comments

Very thoughtful work and thorough revisions. The paper is overall well written, I would still recommend a close proof-reading - there are some in-text citations that are not in alphabetical order, and some changes to the headings during the revisions created format errors. However, at this time I highly recommend the content of the paper. Best wishes on your continued line of inquiry.

·

Basic reporting

The authors tried to conduct a questionnaire survey and related analysis on radiology interns and residents in two hospitals in Saudi Arabia, showing the willingness and subjective initiative of radiology staff with low years of experience in reporting medical adverse events, unprofessional behavior and hidden medical risks, and analyzed the possible reasons that may cause radiologists to be reluctant to report unprofessional behavior seen during routine clinical process. Also discussed possible directions for improvement. The research design of this manuscript is reasonable, the survey objects are representative, and the results are objective and reliable. It has a good guiding significance for the improvement of hospital management measures.After the revision, the content of each part is relatively fuller, and there is a more comprehensive introduction and analysis of the medical training background in Saudi Arabia.

Experimental design

no comment

Validity of the findings

no comment

Additional comments

no comment